# Detection and Classification of Hysteroscopic Images Using Deep Learning

**DOI:** 10.3390/cancers16071315

**Published:** 2024-03-28

**Authors:** Diego Raimondo, Antonio Raffone, Paolo Salucci, Ivano Raimondo, Giampiero Capobianco, Federico Andrea Galatolo, Mario Giovanni Cosimo Antonio Cimino, Antonio Travaglino, Manuela Maletta, Stefano Ferla, Agnese Virgilio, Daniele Neola, Paolo Casadio, Renato Seracchioli

**Affiliations:** 1Division of Gynaecology and Human Reproduction Physiopathology, IRCCS Azienda Ospedaliero-Universitaria di Bologna, 40138 Bologna, Italy; diego.raimondo@aosp.bo.it (D.R.); paolo.casadio@aosp.bo.it (P.C.); renato.seracchioli@aosp.bo.it (R.S.); 2Department of Medical and Surgical Sciences (DIMEC), University of Bologna, 40127 Bologna, Italy; manuela.maletta@studio.unibo.it (M.M.); stefano.ferla2@studio.unibo.it (S.F.);; 3Department of Neuroscience, Reproductive Sciences and Dentistry, School of Medicine, University of Naples Federico II, 80131 Naples, Italy; daniele.neola@unina.it; 4Department of Biomedical Sciences, University of Sassari, 07100 Sassari, Italy; ivano.raimondo@materolbia.com; 5Gynecology and Breast Care Center, Mater Olbia Hospital, 07026 Olbia, Italy; 6Gynecologic and Obstetric Unit, Department of Medical, Surgical and Experimental Sciences, University of Sassari, 07100 Sassari, Italy; capobia@uniss.it; 7Department of Information Engineering, University of Pisa, 56100 Pisa, Italy; federico.galatolo@unipi.it (F.A.G.); mario.cimino@unipi.it (M.G.C.A.C.); 8Unit of Pathology, Department of Medicine and Technological Innovation, University of Insubria, 21100 Varese, Italy; antonio.travaglino@uninsubria.it

**Keywords:** endometrium, uterus, polyps, fibroids, endometrial hyperplasia, endometrial cancer, malignancy, neoplasia, carcinoma, endoscopy, minimally invasive

## Abstract

**Simple Summary:**

This article discusses the potential of deep learning (DL) models in aiding the diagnosis of endometrial pathologies through hysteroscopic images. While hysteroscopy with endometrial biopsy is currently the gold standard for diagnosis, it heavily relies on the expertise of gynecologists. The study aims to develop a DL model for automated detection and classification of endometrial pathologies. Conducted as a monocentric observational retrospective cohort study, it reviewed records and videos of hysteroscopies from patients with confirmed intrauterine lesions. The DL model was trained using these images, with or without incorporating clinical factors. Results indicate that while the DL model showed promising results, its diagnostic performance remained relatively low, even with the inclusion of clinical data. The best performance was achieved when clinical factors were included, with precision, recall, specificity, and F1 scores ranging from 80 to 90% for classification and 85 to 93% for identification tasks. Despite slight improvements in clinical data, further refinement of DL models is warranted for more accurate diagnosis of endometrial pathologies.

**Abstract:**

Background: Although hysteroscopy with endometrial biopsy is the gold standard in the diagnosis of endometrial pathology, the gynecologist experience is crucial for a correct diagnosis. Deep learning (DL), as an artificial intelligence method, might help to overcome this limitation. Unfortunately, only preliminary findings are available, with the absence of studies evaluating the performance of DL models in identifying intrauterine lesions and the possible aid related to the inclusion of clinical factors in the model. Aim: To develop a DL model as an automated tool for detecting and classifying endometrial pathologies from hysteroscopic images. Methods: A monocentric observational retrospective cohort study was performed by reviewing clinical records, electronic databases, and stored videos of hysteroscopies from consecutive patients with pathologically confirmed intrauterine lesions at our Center from January 2021 to May 2021. Retrieved hysteroscopic images were used to build a DL model for the classification and identification of intracavitary uterine lesions with or without the aid of clinical factors. Study outcomes were DL model diagnostic metrics in the classification and identification of intracavitary uterine lesions with and without the aid of clinical factors. Results: We reviewed 1500 images from 266 patients: 186 patients had benign focal lesions, 25 benign diffuse lesions, and 55 preneoplastic/neoplastic lesions. For both the classification and identification tasks, the best performance was achieved with the aid of clinical factors, with an overall precision of 80.11%, recall of 80.11%, specificity of 90.06%, F1 score of 80.11%, and accuracy of 86.74 for the classification task, and overall detection of 85.82%, precision of 93.12%, recall of 91.63%, and an F1 score of 92.37% for the identification task. Conclusion: Our DL model achieved a low diagnostic performance in the detection and classification of intracavitary uterine lesions from hysteroscopic images. Although the best diagnostic performance was obtained with the aid of clinical data, such an improvement was slight.

## 1. Introduction

Hysteroscopy with endometrial biopsy is an endoscopic tool that can be considered the gold standard in the diagnosis of abnormal uterine bleeding (AUB) and endometrial pathology, as it allows the direct visual assessment of endometrium and subsequent histopathological examination [1,2,3,4]. AUB can be caused by benign lesions, such as endometrial polyps, intracavitary myomas, and endometrial hyperplasia without atypias [5,6,7], or pre-malignant and malignant lesions, such as atypical endometrial hyperplasia and endometrial carcinomas [8]. Unfortunately, the experience of the gynecologist plays a crucial role in identifying suspicious areas to be sampled and distinguishing between several endometrial pathologies, with the possibility of failing the correct diagnosis [9].

A valuable help to overcome this limitation could be provided by deep learning (DL), an artificial intelligence (AI) method. AI has recently been introduced in medicine, particularly in disciplines based on the analysis of images, such as pathology, ultrasound, and radiology [10]. For example, AI has shown interesting results in many medical image analysis tasks, such as screening for breast cancer and prediction of lymph node metastasis in cervical cancer [11,12]. In the realm of AI techniques, the utilization of DL for processing and analyzing medical images emerges as highly promising. Deep Convolutional Neural Networks stand as the prevalent DL method for pattern identification in images and videos. Deep Convolutional Neural Networks are able to automatically learn a set of feature detectors, usually over a number of layers (making the model “deep”), from a labeled dataset that “trains” the model to recognize pathologies through image analysis [13,14]. To prepare a DL model for operation, the main dataset is typically divided into two subsets: a training set and a test set. The training set consists of data that are fed into the deep learning network during the iterative training process, known as epochs. Throughout these epochs, the network’s parameters are adjusted to enhance the desired outcome. Following the completion of training, the test dataset is employed to evaluate the performance of the finalized model [15]. DL applications for these tasks may represent a useful tool for clinicians in decision-making and treatment planning [16]. To the best of our knowledge, only two preliminary studies evaluated the performance of DL using hysteroscopy images for diagnosis of benign and malignant endometrial lesions, with favorable results [17,18]. However, none of these studies assessed the performance of DL models in the identification task of intrauterine lesions, as they only reported its accuracy in classifying intrauterine pathologies. In addition, no study evaluated the inclusion of specific clinical factors in the DL model to improve the performance. Moreover, preliminary data on DL performance must be confirmed by different studies before accepting it as a potential clinical aid [19].

In the present study, we aimed to develop a DL model to provide an automated tool for detecting endometrial pathologies and classifying them as benign or malignant intrauterine lesions using hysteroscopic images from a consecutive series of women with pathologically confirmed endometrial lesions.

## 2. Materials and Methods

### 2.1. Study Protocol and Selection Criteria

The study followed an a priori-defined study protocol and was reported according to the Standards for Reporting of Diagnostic Accuracy (STARD) [20]. The study was designed as a monocentric observational retrospective cohort study. 

We reviewed clinical records, electronic databases, and stored videos of hysteroscopies from all consecutive patients with pathological confirmation of intracavitary uterine lesions at IRCCS Azienda Ospedaliero-Universitaria di Bologna, Bologna, Italy, from January 2021 to May 2021. Retrieved hysteroscopic images were used to build a DL model for the classification and identification of intracavitary uterine lesions with and without the aid of clinical factors.

Intracavitary uterine lesions included endometrial polyps, fibroids, endometrial hyperplasia with and without atypia, and endometrial cancer diagnosed at histological examination of hysteroscopic specimens. 

The exclusion criteria were the absence of adequate histological examination, absence of iconographic documentation, presence of uterine dysmorphism, and absence of intrauterine pathology.

### 2.2. Study Outcomes

The primary outcome was the accuracy of the DL model in the classification of intracavitary uterine lesions (overall and by category of lesion) without the aid of specific clinical factors to DL model performance. 

The secondary outcomes were the following: accuracy of the DL model in the classification of intracavitary uterine lesions (overall and by category of lesion) with the aid of specific clinical factors to DL model performance;precision, sensitivity, specificity, and F1 score (i.e., the harmonic mean of precision and sensitivity) of the DL model in the classification of intracavitary uterine lesions (overall and by category of lesion), with and without the aid of specific clinical factors to DL model performance;precision, sensitivity, and F1 score of the DL model in the identification of intracavitary uterine lesions, with and without the aid of specific clinical factors to DL model performance;the best performance of the DL model during testing in the identification and classification of intracavitary uterine lesions (overall and by category of lesion).

Classification refers to the discrimination between three categories of intracavitary uterine lesions: benign focal lesions (i.e., polyps and myomas), benign diffuse lesions (i.e., non-atypical endometrial hyperplasia), and pre-neoplastic/neoplastic lesions (i.e., atypical endometrial hyperplasia and endometrial cancer). Instead, identification referred to the detection of intracavitary uterine lesions. Given the inclusion of only patients with intracavitary uterine lesions diagnosed at histological examination, true negatives were absent for identification metrics. On the other hand, intracavitary uterine lesions of other categories were considered as false negatives for classification metrics. 

Clinical factors assessed for aiding DL model performance were age, menopausal status, AUB, hormonal therapy, and tamoxifen use.

### 2.3. Hysteroscopy and Image Processing

Hysteroscopy with targeted biopsies of intracavitary uterine lesions through 5 French instruments was performed in outpatient settings using 0.9% saline solution distension and a Bettocchi hysteroscope (Karl Storz, Tuttlingen, Germany). Stills and images from hysteroscopic videos of eligible patients were processed for DL model building. Images and videos were captured with two different hysteroscopic systems, one high-definition system and one standard-definition system. Features were extracted from the original image. The system extracts the area of interest for the lesion detected at 224 × 224 pixels required for the classification task. Manual segmentation was performed by an experienced hysteroscopist. 

### 2.4. Deep Learning

We developed an end-to-end DL model for intracavitary uterine lesion identification and classification. The deep learning process comprises three parts: training, validation, and testing. The dataset was divided into three groups at random with a ratio of 60:20:20. Two groups were used for training and validation, and the remaining group was used for testing.

ResNet50 was used as a deep learning model since it can exhibit relatively high accuracy with smaller size datasets and less expensive learning costs. ResNet50 was pre-trained by a million natural images from the Microsoft Common Objects in Context dataset and was fine-tuned using images from the training and validation dataset. 

We used established techniques to reduce over-fitting during the validation process with an iterative method: (a) data augmentation, which is a process synthetically generating additional training examples by using random image transformations; (b) “early stopping”, by which the weights of the network at the point of best performance are saved, as opposed to the weights obtained at the end of training. The performance of the DL model was evaluated using a balanced sampler on image units. 

In our methodology, data augmentation was implemented online, meaning it was applied in real-time during the training of the model. This approach differs significantly from the traditional offline augmentation, where an augmented dataset is prepared in advance before the training process begins. Each training batch underwent a unique set of random transformations, ensuring that the model encountered a diverse range of variations in the training images. This dynamic approach to augmentation is crucial in preventing the model from overfitting, as it learns to generalize better from a constantly varying dataset. The specific augmentation steps included in our process were as follows:Random Vertical and Horizontal Flipping: each image in the training batch had a chance of being flipped either vertically or horizontally. This step introduces a variety of orientations, helping the model to learn features that are orientation-invariant.Random Brightness Adjustment: the brightness of each image was altered using a random factor ranging from 0.8 to 1.2. This variance in brightness ensures the model’s robustness against different lighting conditions.Random Contrast Adjustment: similarly, the contrast of each image was modified with a random factor within the same range (0.8 to 1.2). This step helps in training the model to identify features under various contrast levels.

By incorporating these random transformations, our DL model benefits from a more comprehensive and challenging training environment. This online method of data augmentation plays a significant role in enhancing the model’s ability to accurately classify and identify lesions under diverse imaging conditions, ultimately improving its diagnostic efficacy.

Optimization of hyperparameters was performed using TPESampler as a sample, and SuccessiveHalvingPruner as a pruner, and the train of each set of hyperparameters was replicated 3 times. We used RepeatFactorTrainingSampler with the threshold optimized by hyperparameter optimization. The F1 score average was the optimization metric on the validation set. The hyperparameters are shown in Table 1. Table 2 shows the optimal hyperparameters.

Clinical factors were incorporated into the Region Proposal Network (RPN) and Classification Head and were concatenated to features extracted from the ROI Pooler (Figure 1).

## 3. Results

### 3.1. Study Population and Dataset

During the study period, 703 patients underwent hysteroscopy in our center. Four hundred and thirty-seven were excluded from analysis due to lack of imaging or histological examination or both.

We reviewed a total of 1500 images from 266 patients (image-to-patient ratio = 5.6): 186 (69.92%) patients had benign focal lesions (image-to-patient ratio = 5.97), 25 (9.39%) benign diffuse lesions (image-to-patient ratio = 5.6), and 55 (20.67%) preneoplastic/neoplastic lesions (image-to-patient ratio = 4.55). 

Out of benign focal lesions, 21 were myomas, and 165 were polyps; out of benign diffuse lesions, 19 were polypoid endometrium, and 6 were endometrial hyperplasia without atypia; out of preneoplastic and neoplastic lesions, 7 were atypical endometrial hyperplasia, 12 were endometrial intraepithelial neoplasia, and 36 were endometrial cancers.

Clinical data about the whole study population and by category of intracavitary uterine lesions are summarized in Table 3. Patients were randomly included in the training (*n* = 157), validation (*n* = 54), and testing (*n* = 55) cohorts (Table 4).

### 3.2. Model Performance

Overall, the accuracy of the model in classifying uterine intracavitary lesions without the aid of specific clinical factors was 85.09 ± 1.18%. Specifically, such accuracy was 79.55 ± 1.29% for benign focal lesions, 90.1 ± 0.91% for benign diffuse lesions, and 85.63 ± 1.16% for malignant lesions.

Table 5 and Table 6 show the accuracy, precision, sensitivity, specificity, and F1 score of the DL model in the classification of intracavitary uterine lesions, without and with the aid of specific clinical factors, to DL model performance, respectively.

Table 7 shows the precision, sensitivity, and F1 score of the DL model in the identification of intracavitary uterine lesions, with and without the aid of specific clinical factors, to DL model performance.

For the classification task, the best performance was achieved in all the categories with the aid of clinical factors, as shown in Table 8.

For the identification task, the best performance was achieved with the aid of clinical factors with detection of 85.82%, precision of 93.12%, recall of 91.63%, and an F1 score of 92.37%.

## 4. Discussion

This study showed that the DL model had low overall accuracy in the detection and classification of uterine intracavitary diseases. The best performance of the DL model was obtained with the aid of clinical factors for both tasks. However, such an improvement was slight. 

Although hysteroscopy with endometrial biopsy appears as the gold standard diagnostic tool for AUB and uterine intracavitary diseases [22], it is affected by operator experience in detecting suspicious areas to be sampled and distinguishing between several diseases. Moreover, hysteroscopic diagnosis of uterine intracavitary diseases can be challenging even if it is performed by expert operators [23]. Hysteroscopy has shown a low sensitivity especially for endometrial hyperplasia since such disease may not show evident hysteroscopic signs, simulating a second-phase or dysfunctional endometrium, or endometrial polyps [1,2,3,4,9,24].

Recently, some studies have attempted to build DL models to try to overcome these limitations. Takahashi et al. have recently employed DL models on 177 patients with AUB in order to increase the hysteroscopy accuracy in cancer diagnosis [17]. In detail, the Takahashi DL model distinguished atypical endometrial hyperplasia and endometrial cancer from polyps, fibroids, or normal endometrium with a 90% accuracy. However, this study might be affected by several limitations: (i) it did not evaluate the ability of the DL model in detection of endometrial lesions; (ii) it did not assess the possible aid of clinical factors on machine learning performance; (iii) it did not include cases with non-atypical endometrial hyperplasia; (iv) it did not evaluate histology as a reference standard for all cases; (v) it used a dataset with images from only one hysteroscopic system, limiting the generalizability of the findings.

Yet, Zhang et al. have built a DL model on 454 patients with histologically confirmed intracavitary lesions, showing an overall accuracy of up to 80.8% and 90% in correctly classifying lesions as benign or premalignant/malignant, respectively. However, also this study did not evaluate DL model accuracy in the detection of endometrial lesions and possible improvement in accuracy with the aid of clinical factors [18].

Zhao et al. developed a DL model to automatically detect only endometrial polyps in real-time hysteroscopic videos with an accuracy of up to 95%; unfortunately, they did not perform any classification of the lesions [25].

None of these studies used a DL model to identify and classify intracavitary uterine lesions at the same time. Therefore, we built a DL model for these purposes and evaluated its diagnostic performance (identification and classification) on hysteroscopy images from women performing the exam for AUB or sonographic suspect of an intrauterine lesion, then confirmed at pathological examination [26]. To the best of our knowledge, our study may be the first study with these aims and study population in the literature. Furthermore, our DL model may be the first one to include the aid of clinical factors in the field.

As previously stated, in the present study, our DL model showed a low accuracy in the detection and classification of intracavitary diseases. This observation may reflect the heterogeneity of uterine intracavitary pathology, the small size, and the heterogeneity of the dataset. Moreover, the lack of images of normal cavities and the small number of patients led to a dataset imbalance problem. 

Anyway, the best performance of our DL model is close to that of the above-mentioned larger studies. Our DL model might be an updated starting point for future improved DL models in the field.

In order to improve the diagnostic performance of the DL model in the detection and classification of intrauterine lesions, future research should be focused on specific training of the DL machine on the detection between normal and abnormal cavities and recognition of each category with a larger and balanced dataset including high-definition images and videos. After DL model building, the model should undergo external validation and improvement, with the inclusion of further images and videos from other centers. When a high DL model performance is obtained, the inclusion of cases with other rarer intrauterine pathologies (e.g., Mullerian malformations, atypical polypoid adenomioma, pecoma, sarcoma, trophoblastic disease, retained products of conception) [27,28,29,30] might make the DL model testable in the clinical practice thorough comparison of diagnostic performance by expert endoscopists.

## 5. Conclusions

In this study, our DL model achieved a low diagnostic performance in the detection and classification of intracavitary uterine lesions from hysteroscopic images. Although the best diagnostic performance was obtained with the aid of clinical data, such an improvement was slight. However, our DL model might be an updated starting point for future improved DL models in the field based on larger datasets.

Our study underscores the importance of continued research in refining DL models for uterine lesion detection and classification. Future efforts should prioritize the expansion of datasets with high-definition images, the inclusion of diverse uterine pathologies, and external validation across multiple centers. Moreover, the addition of normal uterine cavity images and rarer intrauterine lesions to the training set might allow to enhance the DL model’s diagnostic accuracy.

In conclusion, while our DL model represents a promising step towards automated uterine lesion diagnosis, further refinement and validation is needed before its integration into clinical practice. By addressing current limitations and leveraging advances in AI technology, future DL models hold the potential to significantly improve the accuracy and efficiency of uterine pathology diagnosis, ultimately benefiting patient care and outcomes.

## Figures and Tables

**Figure 1 cancers-16-01315-f001:**
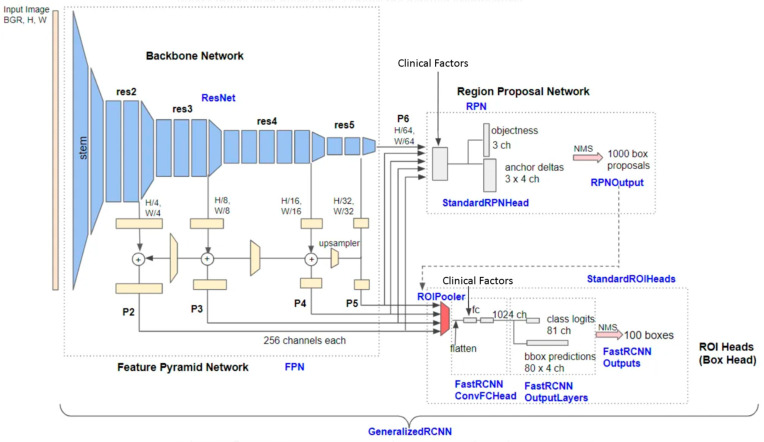
Detailed architecture of Base-RCNN-FPN (adapted from H. Honda [21]).

**Table 1 cancers-16-01315-t001:** Hyperparameters.

Hyperparameter	Sampling Method	Range/Options
Learning Rate (lr)	Log Uniform Distribution	1 × 10^−5^ to 1 × 10^−2^
RPN Loss Weight	Uniform Distribution	0 to 1
ROI Heads Loss Weight	Uniform Distribution	0 to 1
ROIs Per Image	Categorical	32, 64, 128, 256, 512
Random Brightness *	Uniform Distribution	0 to 1
Random Contrast *	Uniform Distribution	0 to 1
Repeat Factor Th **	Uniform Distribution	0.1 to 1

* related to data augmentation. ** minority class repetition factor.

**Table 2 cancers-16-01315-t002:** Optimal Hyperparameters.

Hyperparameter	Value
Learning Rate (lr)	0.0015884830145038431
ROIs Per Image	256
RPN Loss Weight	0.8635956597511065
ROI Heads Loss Weight	0.5995106068965408
Repeat Factor Th **	0.45776224748623207
Random Contrast *	0.2
Random Brightness *	0.2

* related to data augmentation. ** minority class repetition factor.

**Table 3 cancers-16-01315-t003:** Clinical data on the whole study population and category of intracavitary uterine lesions.

	Patients (*n* = 266)	Benign Focal Lesions (*n* = 186)	Benign Diffuse Lesions (*n* = 25)	Preneoplastic and Neoplastic Lesions (*n* = 55)
Age, mean (range)	53.5 (27–87)	52 (27–83)	45 (29–76)	62.2 (39–87)
Menopausal status, *n* (%)	132 (49.62)	83 (44.62)	5 (20)	44 (80)
Abnormal uterine bleeding, *n* (%)	118 (44.36)	69 (37.09)	7 (28)	42 (76.3)
Hormonal therapy, *n*(%)	24 (9.02)	13 (6.98)	0 (0)	11 (20)
Tamoxifen users, *n* (%)	6 (2.25)	5 (2.68)	1 (4)	0 (0)

**Table 4 cancers-16-01315-t004:** Characteristics of the dataset.

	Patients (*n*)	Images (*n*)	Patients in Training Set (*n*)	Images in Training Set (*n*)	Patients in Validation Set (*n*)	Images in Validation Set (*n*)	Patients in Testing Set (*n*)	Images in Testing Set (*n*)
Benign focal lesion	186	1110	111	667	37	273	38	170
Benign diffuse lesion	25	140	14	82	7	38	7	20
Preneoplastic and neoplastic lesion	55	250	32	159	10	35	10	56
Total	266	1500	157	908	54	355	55	237

**Table 5 cancers-16-01315-t005:** Accuracy, precision, sensitivity, specificity, and F1 score of the DL model in the classification of intracavitary uterine lesions without clinical data. Values are expressed as % (95% CI).

	Precision	Recall	Specificity	F1	Accuracy
Benign focal lesion	82.96 ± 0.54	92.64 ± 2.14	36.85 ± 7.18	87.29 ± 0.92	79.55 ± 1.29
Benign diffuse lesion	29.93 ± 8.58	21.17 ± 5.83	97.13 ± 1.45	28.27 ± 4.02	90.1 ± 0.91
Pre-neoplastic/neoplastic lesion	51.7 ± 6.64	35.16 ± 7.67	94.32 ± 1.81	42.19 ± 5.32	85.63 ± 1.16
Overall	63.03 ± 6.14	49.66 ± 5.5	76.1 ± 3.67	52.58 ± 3.43	85.09 ± 1.18

**Table 6 cancers-16-01315-t006:** Accuracy, precision, sensitivity, specificity, and F1 score of the DL model in the classification of intracavitary uterine lesions with clinical data. Values are expressed as % (95% CI).

	Precision	Recall	Specificity	F1	Accuracy
Benign focal lesion	84.25 ± 1.18	94.31 ± 2.24	39.59 ± 6.79	88.8 ± 0.97	81.97 ± 1.15
Benign diffuse lesion	48.78 ± 6.22	29.92 ± 5.99	96.2 ± 1.45	34.45 ± 4.65	90.61 ± 1.14
Pre-neoplastic/neoplastic lesion	67.97 ± 5.51	32.19 ± 7.06	96.52 ± 1.35	43.01 ± 5.43	87.07 ± 1
Overall	67 ± 4.4	52.14 ± 5.37	77.44 ± 3.37	55.42 ± 3.76	86.55 ± 1.15

**Table 7 cancers-16-01315-t007:** Precision, sensitivity, and F1 score of the DL model in the identification of intracavitary uterine lesions, with and without the aid of specific clinical factors, to DL model performance. Values are expressed as % (95% CI).

Clinical Factors	Detection	Precision	Recall	F1
No	66.41 ± 3.39	88.27 ± 2.54	72.87 ± 3.5	79.43 ± 2.55
Yes	66.58 ± 4.64	86.82 ± 3.34	73.49 ± 4.56	79.18 ± 3.62

**Table 8 cancers-16-01315-t008:** Best performance of the DL Model in the classification task. Values are expressed as %.

Lesion	Precision	Recall	Specificity	F1	Accuracy
Benign focal lesion	85.23	94.07	46.34	89.44	82.95
Benign diffuse lesion	37.5	50	93.9	42.86	90.91
Pre-neoplastic/neoplastic lesion	72.73	27.59	97.96	40	86.36
Overall	80.11	80.11	90.06	80.11	86.74

## Data Availability

The data that support the findings of this study are available on request from the corresponding author.

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
