# Peer review of "Detection and Classification of Hysteroscopic Images Using Deep Learning"

_cancers, 2024, doi:10.3390/cancers16071315_

Round 1

Reviewer 1 Report

Comments and Suggestions for Authors

Congratulations to the authors for this piece of research. I enjoyed reading the work.

Authors should see the comments below to help improve the work.

1.        The simple summary section and the abstract contain very similar content that is often repeated. I suggest the authors should merge these sections.

2.        The introduction section needs a lot of improvement. Authors should include more sources that describe the disease and discuss DL models in relative detail.

3.        The authors should also consider restructuring or renaming some of the sections for example, sections 2.2 and 3.3 have the same heading title. How many study outcomes are being referred to?

4.        What parameters have been used to tune the ResNet50 model for training? What are the values considered for learning rate, batch normalisation, etc?

5.        How have the clinical factors helped in aiding the DL model’s performance? How have you incorporated these factors into ResNet50 as it uses images as the sole input to the model?

6.        In line 180, a total of 1,500 images were obtained from 266 patients. What image processing have you done? What is the ratio of the images to patients?

7.        Also, 70% of the images belong to the focal lesions class. This was apparent with the specificity of the model as it is obvious that the model is biased towards this class. Authors should consider using generator models or other techniques to deal with class imbalance.

8.  The Conclusion sectionshould be expounded a bit more. 

Reviewer 2 Report

Comments and Suggestions for Authors

The article titled "Detection and Classification of Hysteroscopic Images Using Deep Learning" discusses the development of a deep learning (DL) model to assist in diagnosing endometrial pathologies through hysteroscopic images. The study is a monocentric observational retrospective cohort study that reviewed records and videos of hysteroscopies from patients with confirmed intrauterine lesions. The DL model was trained using these images, with or without incorporating clinical factors, and the study aimed to evaluate the model's performance in identifying and classifying these lesions.

The shortcomings of the article, based on the provided information, include:

1. Limited Diagnostic Performance: The article acknowledges that the DL model achieved a low diagnostic performance in detecting and classifying intracavitary uterine lesions from hysteroscopic images. This suggests that the model may still need to be more reliable for clinical use with further refinement.

2. Slight Improvement with Clinical Data: Although the best diagnostic performance was obtained with the aid of clinical data, the improvement was described as slight. This indicates that the inclusion of clinical factors did not significantly enhance the model's accuracy, which could be a limitation in the practical application of the model.

3. Retrospective and Monocentric Nature: The study is retrospective and was conducted at a single center, which may limit the generalizability of the findings. A multicentric study with a prospective design might provide more robust and generalizable results.

4. Absence of True Negatives in Identification Metrics: The study included only patients with intracavitary uterine lesions diagnosed at histological examination, which means true negatives were absent for identification metrics. This could affect the assessment of the model's performance in real-world scenarios where true negatives are present.

5. Potential Overfitting: Despite using established techniques to reduce overfitting, such as data augmentation and early stopping, the risk of overfitting must be considered entirely, especially given the complex nature of medical image analysis and the relatively small dataset size.

6. Dataset Size and Diversity: The study reviewed 1,500 images from 266 patients, which may need to be revised to capture the total variability of endometrial pathologies. A more extensive and diverse dataset could improve the model's performance and robustness.

7. Resolution Reduction: Images were reduced to a resolution of $$224 \times 224$$ pixels for model training. This reduction in resolution might lead to the loss of essential details crucial for accurate lesion identification and classification.

8. Manual Segmentation: The manual segmentation performed by an experienced hysteroscopist introduces a subjective element to the data preparation process, which could influence the training and performance of the DL model.

9. Need for External Validation: The article suggests that different studies must confirm preliminary data on DL performance before it can be accepted as a potential clinical aid. This indicates a need for external validation of the model's performance across different populations and clinical settings.

10. Specificity of Some Lesions: The specificity for benign focal lesions was relatively low at 36.85, which could indicate a high rate of false positives for this category. This is a significant shortcoming, as it could lead to unnecessary interventions or anxiety for patients.

These shortcomings highlight the need for further research and development to improve the accuracy and reliability of DL models for diagnosing endometrial pathologies using hysteroscopic images.

Comments on the Quality of English Language

minor

Round 2

Reviewer 1 Report

Comments and Suggestions for Authors

Thanks for providing the revised version of this work.

However, further clarifications need to be provided for the image input. You have stated a patient-to-image ratio of 5.6, but this is not the case for the  Benign Focal lesion and Preneoplastic and neoplastic classes. 

You have also stated that you haven't preprocessed the images. How did you get 1110 images from 186 patients for the benign focal lesion? Is this the recording of the images? this is not 1:1, meaning some patients will have more images than others in the population sample. So what is the rationale for this duplication? are the images per patient different?

Thanks

Author Response

RE: cancers-2904032

Dear Editor:

Thank you for giving us the chance to enhance our manuscript Detection And Classification of Hysteroscopic Images Using Deep Learning.

Below is each question raised by the Reviewers, followed by our response, as well as the position in the paper where issue is mentioned.

We submitted the revised manuscript using the “track changes” feature. We also submitted a clean, non-edited final version of the revised paper. Reported lines refer to the clean version of the manuscript.

REVIEWER #1

Comment #1

  1. A) Thanks for providing the revised version of this work.

However, further clarifications need to be provided for the image input. You have stated a patient-to-image ratio of 5.6, but this is not the case for the Benign Focal lesion and Preneoplastic and neoplastic classes. 

You have also stated that you haven't preprocessed the images. How did you get 1110 images from 186 patients for the benign focal lesion? Is this the recording of the images? this is not 1:1, meaning some patients will have more images than others in the population sample. So what is the rationale for this duplication? are the images per patient different?

  1. B) Response: We thank the Reviewer for the comment. In fact, we reported the total patient-to-images ratio=5.6. However, the ratio is different for each type of endometrial lesion, as reported in Table 4. We reported the specific ratios in the revised manuscript, as requested.

For what concerns the number of images, we stated in the Methods section that we retrospectively extracted the images for our model from images and videos of hysteroscopic exams. In particular, from each hysteroscopic video several frames (then images) were extracted, each image containing different information for our model. Therefore, more than one image per patient should not be considered as duplicated, but different views of a lesion, which provide different information to the AI model.

  1. C) Location: Lines 191-193; Table 4

Thank you and we look forward to hearing from you.

Sincerely,

Antonio Raffone, M.D., (for all authors)

Reviewer 2 Report

Comments and Suggestions for Authors

thank you for the revisions.

i am satisfied with the revisions

Author Response

We thank the reviewer for the kind comment.